# New Phosphonite Ligands with High Steric Demand and Low Basicity: Synthesis, Structural Properties and Cyclometalated Complexes of Pt(II)

**María M. Alcaide [1], Matteo Pugliesi [2], Eleuterio Álvarez [1], Joaquín López-Serrano [1,*] and Riccardo Peloso [1,*]**

[1] Instituto de Investigaciones Químicas (IIQ), Departamento de Química Inorgánica and Centro de Innovación en Química Avanzada (ORFEO−CINQA), Consejo Superior de Investigaciones Científicas (CSIC), Universidad de Sevilla, 41092 Sevilla, Spain; mmalcaide@us.es (M.M.A.); ealvarez@iiq.csic.es (E.Á.)

[2] Dipartimento di Chimica e Chimica Industriale, Universitá di Pisa, Via G. Moruzzi 13, I-56124 Pisa, Italy; matteo.pugliesi@iccom.cnr.it

[*] Correspondence: joaquin.lopez@iiq.csic.es (J.L.-S.); rpeloso@us.es (R.P.)

**Abstract:** Two phosphonite ligands bearing the highly sterically demanding 2,6-bis (2,6-dimethylphenyl)phenyl group (Ar$^{Xyl_2}$), **PAr$^{Xyl_2}$(OPh$^{NO_2}$)$_2$** and **PAr$^{Xyl_2}$(OPh$^{NO_2,Me}$)$_2$**, were prepared from the parent dihalophosphines PAr$^{Xyl_2}$X$_2$ (X = Cl, Br) and the corresponding phenols, 4-nitrophenol and 4-nitro-2,6-dimethylphenol, respectively. DFT methods were used to examine their structural features and to determine three steric descriptors, namely the Tolman cone angle, the percentage of buried volume, and the percentage of the coordination sphere protected by the ligand. A comparison with the related terphenyl phosphines is also provided. Reactions of **PAr$^{Xyl_2}$(OPh$^{NO_2}$)$_2$** and **PAr$^{Xyl_2}$(OPh$^{NO_2,Me}$)$_2$** with several Pt(II) precursors were investigated, revealing a high tendency of both phosphonites to undergo C-H activation processes and generate five- or six-membered cyclometalated structures. The coordination chemistry of the new ligands was explored with isolation, among others, of three carbonyl complexes, **1-3·CO**, and the triphenylphosphine adduct **3·PPh$_3$**. X-ray diffraction methods permitted the determination of the solid-state structures of the mononuclear methyl carbonyl complex **1·CO**, the dinuclear chloride-bridged complex **2** and the doubly cyclometalated complex **3·SMe$_2$**, including the conformations adopted by the ligands upon coordination. All of the new compounds were characterized by multinuclear NMR spectroscopy in solution.

**Keywords:** P(III) ligands; phosphonites; platinum complexes; cyclometalated structures; DFT; bulky ligands

## 1. Introduction

In spite of the long and rich history of phosphorus donor ligands in coordination chemistry [1–7], the development of new molecules containing trivalent phosphorus atoms capable of stabilizing diverse types of metal complexes [8–13], supramolecular structures [14], and nanoparticles [11,15,16] is still an area of intense research activity. Electronic and steric properties of this class of Lewis bases can easily be modulated with the introduction of atoms or molecular fragments of the desired characteristics directly at the phosphorus atom, or in selected positions of the organic groups that are usually bonded to it. In addition to the widespread use of P(III) ligands in catalytic processes [5,6], bulky phosphines and related mono- or polydentate ligands have permitted the stabilization of particularly elusive metal complexes, such as the first σ-methane complexes of Rh(I) characterized by Brookhart and collaborators taking advantage of the steric protections provided by the hybrid bis (phosphinite)-pyridine ligand, 2,6-(*t*Bu$_2$PO)$_2$C$_5$H$_3$N [17]. In recent years, our research group described the synthesis of a number of highly steric demanding tertiary phosphines bearing terphenyl groups [18,19] and disclosed a variety of stoichiometric [20–25] and

catalytic reactions [21,26–28] involving their complexes with late transition metals. Closely related to our findings, Nicasio and coworkers successfully employed dialkyl terphenyl phosphines to stabilize rare examples of 2-coordinate complexes of group 10 metals in the zero-oxidation state [29].

In contrast to the increasing data available on terphenyl phosphines and their applications, terphenyl-substituted phosphite and phosphinite have not been explored yet, with the sole exception of a ferrocene-based terphenyl phosphonite reported recently by Breher, together with a gold(I) complex used in hydroamination catalysis [30]. In this context, we became interested in the preparation of terphenyl phosphonites of the general formula Ar′P(OR)$_2$ (Ar′ = $m$-terphenyl) with special emphasis on the development of sterically hindered $\pi$-accepting ligands. The conjunction of these characteristics is expected to be valuable for the stabilization of coordinatively unsaturated metal complexes in low-oxidation states [21,29,31] and, in general, in enhancing the electrophilicity of their coordination compounds.

In the present work, we report on the synthesis and characterization of two phosphonite ligands bearing the bulky Ar$^{Xyl_2}$ fragment, **PAr$^{Xyl_2}$(OPh$^{NO_2}$)$_2$** and **PAr$^{Xyl_2}$(OPh$^{NO_2,Me}$)$_2$**, and the electron-poor 4-nitrophenoxy and 4-nitro-2,6-dimethylphenoxy group, respectively, together with a theoretical evaluation of their steric properties. Moreover, we describe their reactions with several Pt(II) precursors, which give access to a series of mononuclear and dinuclear cyclometalated complexes.

## 2. Results

### 2.1. Synthesis and Properties of Terphenyl Phosphonites

The synthesis of the phosphonite ligands **PAr$^{Xyl_2}$(OPh$^{NO_2}$)$_2$** and **PAr$^{Xyl_2}$(OPh$^{NO_2,Me}$)$_2$** was selectively accomplished with high yields by reacting the parent dihalo phosphines PAr$^{Xyl_2}$X$_2$ (X = Cl, Br; Ar$^{Xyl_2}$ = 2,6-bis(2,6-dimethylphenyl)phenyl) with 4-nitrophenol and 4-nitro-2,6-dimethylphenol, respectively, in the presence of triethylamine (Scheme 1) [30,32]. Both compounds were isolated as yellow solid materials and feature high solubility in common organic solvents, including saturated hydrocarbons. Although they can be stored indefinitely under nitrogen or argon without decomposition and can be exposed to the air for a short time, phosphonites **PAr$^{Xyl_2}$(OPh$^{NO_2}$)$_2$** and **PAr$^{Xyl_2}$(OPh$^{NO_2,Me}$)$_2$** are slowly hydrolyzed by moisture both in solution and in the solid state, affording the corresponding monosubstituted phosphinic acids [33].

**Scheme 1.** Synthesis of the terphenyl phosphonite ligands, **PAr$^{Xyl_2}$(OPh$^{NO_2}$)$_2$** and **PAr$^{Xyl_2}$(OPh$^{NO_2,Me}$)$_2$**.

Phosphonites **PAr$^{Xyl_2}$(OPh$^{NO_2}$)$_2$** and **PAr$^{Xyl_2}$(OPh$^{NO_2,Me}$)$_2$** were characterized by multinuclear NMR spectroscopy. Their $^{31}$P{$^1$H} NMR spectra in CDCl$_3$ consist of a unique singlet at ca. 165 ppm, in the expected range for this type of P(III) compounds [34]. In the $^1$H NMR of both compounds, the terphenyl fragment gives rise to a pattern of signals that can be rationalized by an apparent $C_{2v}$ symmetry with the $C_2$ axis containing the P—C

bond. So, only one resonance at ca. 2.1 ppm is observed for the four methyl groups of the xylyl fragments and only two sets of signals for the aromatic protons. On the other hand, whereas the *ortho* and *meta* protons of the **OPh$^{NO_2}$** groups in **PAr$^{Xyl_2}$(OPh$^{NO_2}$)$_2$** only originate two doublets as in the free 4-nitrophenol, the introduction of two methyl groups at the *ortho* positions impedes the rotation of the phenoxy moieties on the NMR time scale, which results in two clearly differentiated signals for the benzylic protons of the **OPh$^{NO_2,Me}$** group. Schematic stereo views of the molecular structures of the two phosphonite ligands with rotations accounting for the apparent symmetry in solution are shown in Figure 1.

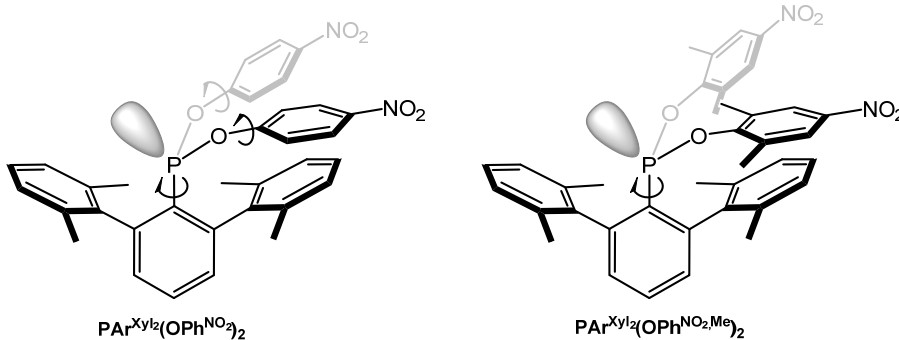

**Figure 1.** Schematic views of the 3D structures of **PAr$^{Xyl_2}$(OPh$^{NO_2}$)$_2$** and **PAr$^{Xyl_2}$(OPh$^{NO_2,Me}$)$_2$**. Curved arrows indicate fast rotations which account for the apparent symmetry observed in NMR experiments.

In order to compare the conformation adopted by these phosphonites with those of the related terphenyl phosphines, the gas-phase structures of both phosphonites were calculated by DFT methods (ωB97XD/6-31g(d,p) level; see below) and are shown in Figure 2.

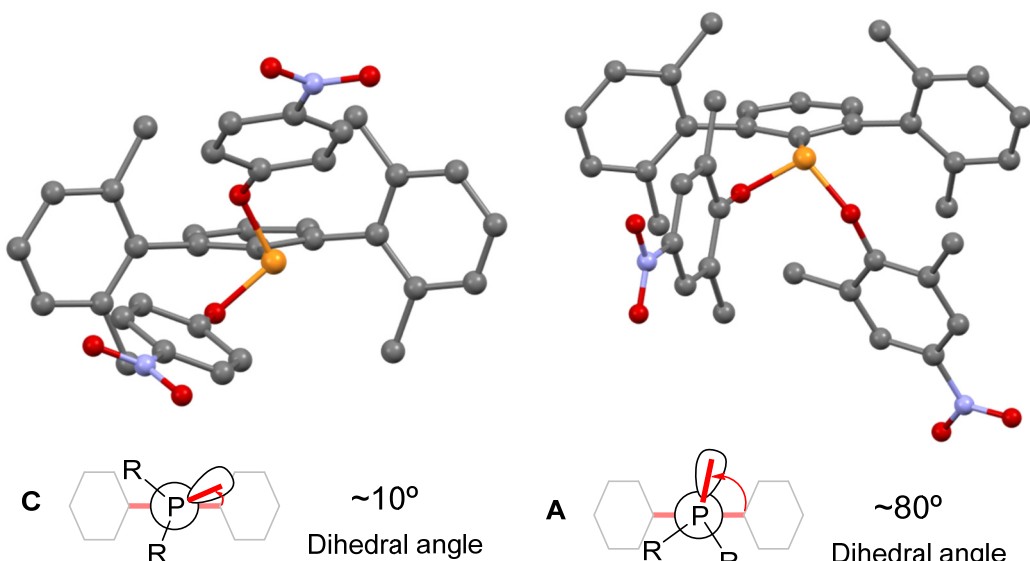

**Figure 2.** Views of the calculated gas-phase structures of PAr$^{Xyl_2}$(OPh$^{NO_2}$)$_2$ (**left**) and PAr$^{Xyl_2}$(OPh$^{NO_2,Me}$)$_2$ (**right**) (C: grey; N: purple; O: red; P: orange). Conformations C and A relate to previously reported discussions on terphenyl phosphines ligands [18,19,26]. The two planes defining the dihedral angle are that containing the central ring of the terphenyl group and that containing the P—C bond and the axis of the P lone pair.

Curiously, the bulkier phosphonite, **PAr$^{Xyl_2}$(OPh$^{NO_2,Me}$)$_2$**, shows the same conformation adopted by the least sterically demanding terphenyl phosphines of the series that we have recently examined [18], namely PMe$_2$Ar$^{Dtbp_2}$ (Ar$^{Dtbp_2}$ = 2,6-(3,5-$t$Bu$_2$C$_6$H$_3$)C$_6$H$_3$) and PMe$_2$Ar$^{Ph_2}$ (Ar$^{Ph_2}$ = 2,6-Ph$_2$C$_6$H$_3$), the two featuring P-Me bonds and no ring substitution

at positions 2 and 6 of the flanking rings. Conversely, the less hindered phosphonite, **PAr$^{\text{Xyl}_2}$(OPh$^{\text{NO}_2}$)$_2$**, adopts the conformation displayed by highly steric demanding phosphines, such as P*i*Pr$_2$Ar$^{\text{Xyl}_2}$ and PAr$^{\text{Xyl}_2}$Cy$_2$. These findings show that the introduction of OAr substituents on the P atom significantly changes not only the electron-donating properties of this class of ligands, but also their conformations. It is important to mention that for both phosphonites' ligands, the calculated C-O-P angles are much larger than expected for a $sp^3$ oxygen atom and span from ca. 120° to ca. 150°, the latter in **PAr$^{\text{Xyl}_2}$(OPh$^{\text{NO}_2,\text{Me}}$)$_2$**, thus providing significant relief to the intramolecular steric repulsions between the substituted phenoxy fragments and xylyl rings.

To provide a detailed description of the steric properties of **PAr$^{\text{Xyl}_2}$(OPh$^{\text{NO}_2}$)$_2$** and **PAr$^{\text{Xyl}_2}$(OPh$^{\text{NO}_2,\text{Me}}$)$_2$**, we decided to calculate three steric parameters: the Tolman cone angle (TCA), the percentage of buried volume (%Vbur), and the percentage of the coordination sphere protected by the ligand (%G). For the sake of comparison, a less hindered phosphonite, PAr$^{\text{Xyl}_2}$(OMe)$_2$, was also examined using its DFT-optimized geometry (Table 1).

**Table 1.** %V$_{\text{bur}}$, TCA y %G values for the phosphonites studied in this work and for some related phosphine ligands.

| Ligand | %Vbur | TCA(°) | %G | Ref |
|---|---|---|---|---|
| **PAr$^{\text{Xyl}_2}$(OPh$^{\text{NO}_2}$)$_2$** | 59.7 | 218.1 | 57.2 | this work |
| **PAr$^{\text{Xyl}_2}$(OPh$^{\text{NO}_2,\text{Me}}$)$_2$** | 55.3 | 226.5 | 68.2 | this work |
| **PAr$^{\text{Xyl}_2}$(OMe)$_2$** | 49.1 | 210.8 | 50.3 | this work |
| PAr$^{\text{Xyl}_2}$Me$_2$ | 38.6 | 153.9 | 38.8 | [18] |
| PAr$^{\text{Xyl}_2}$(*c*-C$_6$H$_{11}$)$_2$ | 45.2 | 187.3 | 47.1 | [18] |
| PAr$^{\text{Dtbp}_2}$(*c*-C$_5$H$_9$)$_2$ | 56.7 | 212.4 | 46.2 | [18] |

It is pertinent to compare the steric parameters calculated for the phosphonites of this work to those of related terphenyl phosphines, which have recently been reported by some of us. On the basis of the following observations and data:

- The highest TCA value among the phosphine series is 212.4°, for PAr$^{\text{Dtbp}_2}$(*c*-C$_5$H$_9$)$_2$, in which the flanking rings of the terphenyl moiety are functionalized with two bulky *tert*-butyl groups at the *meta* positions and two cyclopentyl groups are bound to the P atom;
- The TCA values for PAr$^{\text{Xyl}_2}$R$_2$ phosphines, which bear the same terphenyl group as the phosphonites of this work, are in the range 153.9–187.3, with the highest value for the highly steric demanding R = cyclohexyl;
- **PAr$^{\text{Xyl}_2}$(OPh$^{\text{NO}_2}$)$_2$** and **PAr$^{\text{Xyl}_2}$(OPh$^{\text{NO}_2,\text{Me}}$)$_2$** have TCA values of 218.1 and 226.5, respectively, i.e., significantly higher than those of the related PR$_2$Ar$^{\text{Xyl}_2}$;
- The TCA increases by ca. 60° from the dimethyl terphenyl phosphine PAr$^{\text{Xyl}_2}$Me$_2$ to the dimethoxy phosphonite **PAr$^{\text{Xyl}_2}$(OMe)$_2$**;
- Data reported for a series of PR$_3$ and P(OR)$_3$ ligands indicate that, in most cases, the TCA is reduced slighlty when passing from a phosphane to the related phosphite, e.g., from PPh$_3$ (TCA$_{\text{tetrehedral}}$ = 165.8°) to P(OPh)$_3$ (TCA$_{\text{tetrehedral}}$ = 152.9°).

We could conclude that the introduction of substituted phenoxy groups at the phosphorus atom, in conjunction with the terphenyl group, significantly increases the cone angles of this class of ligands, which is likely a consequence of the flexibility of the POC angle (*vide supra*). So, the introduction of O spacers between the P atom and the R groups relieves the repulsion of the latter with the Xyl rings but drives them closer to the metal centre, thus increasing the TCA. Accordingly, terphenyl phosphonites feature higher %V$_{\text{bur}}$ values than the related phosphines and provide, therefore, steric protection closer to the metal center. Finally, when comparing the characteristics of **PAr$^{\text{Xyl}_2}$(OPh$^{\text{NO}_2}$)$_2$** and **PAr$^{\text{Xyl}_2}$(OPh$^{\text{NO}_2,\text{Me}}$)$_2$**, it can be noted that the introduction of four methyl groups at the *ortho* positions of the phenoxyde rings results in increased TCA and %G values and a slightly reduced %V$_{\text{bur}}$ value. As a final comment about the quantification of the steric properties of terphenyl-substituted

P(III) ligands, it is convenient to stress that the method used for this purpose does not consider that, upon coordination, the conformation of the ligand, and particularly the orientation of the P lone pair, may change with parallel variation in the steric descriptors.

*2.2. Synthesis and Characterization of Cyclometalated Complexes of Pt(II) with PAr$^{Xyl_2}$(OPh$^{NO_2}$)$_2$ and PAr$^{Xyl_2}$(OPh$^{NO_2,Me}$)$_2$*

Platinum complexes bearing terphenyl phosphines have recently proved to possess an interesting structural chemistry, unprecedented reactivity, and also attractive catalytic properties. On these grounds, we decided to undertake analogous studies with the new phosphonite ligands of this work, which were expected to enhance the electrophilic character of the metal center owing to their reduced electron-donating ability, while maintaining the high steric protection provided by the terphenyl group.

In our first experiments, typical platinum(II) dichloride precursors, namely PtCl$_2$(DMSO)$_2$, PtCl$_2$(1,5-cyclooctadiene), and PtCl$_2$(SMe$_2$)$_2$, were treated at room temperature with equimolar amounts of ligands **PAr$^{Xyl_2}$(OPh$^{NO_2}$)$_2$** or **PAr$^{Xyl_2}$(OPh$^{NO_2,Me}$)$_2$** but no reaction was observed, either at higher temperatures (up to 100 °C) or using more polar organic solvents. We thought that the low trans-effect of the chloride was not sufficient to promote the substitution of the neutral ligands of the precursor with the bulky and poorly donating phosphonites. Consequently, the more reactive dinuclear precursor [PtMe$_2$(μ-SMe$_2$)]$_2$ was assayed in a 1:2 reaction with **PAr$^{Xyl_2}$(OPh$^{NO_2}$)$_2$** at room temperature in dichloromethane (Scheme 2). Analysis of the reaction mixture by $^{31}$P{$^1$H} NMR spectroscopy revealed the quantitative and selective formation of a new Pt complex after a few hours, as inferred by the disappearance of the singlet resonance of the ligand at ca. 166 ppm and the formation of a new one with satellites at 169.3 (s, $^1J_{PPt}$ = 2550 Hz). The new complex was isolated as a colorless, air-stable, solid material and characterized by $^1$H and $^{13}$C{$^1$H} NMR, which allowed us to propose the cyclometalated structure **1·SMe$_2$** depicted in Scheme 2.

**Scheme 2.** Synthesis of complexes **1** and **2**.

Particularly informative were the following NMR data: (a) $^1$H and $^{13}$C{$^1$H} NMR spectra of **1·SMe$_2$** contain one single Pt-CH$_3$ signal, at 0.43 ($^3J_{HP}$ = 8.6 Hz, $^2J_{HPt}$ = 65.1 Hz) and 8.9 ($^1J_{CPt}$ = 145 Hz) ppm, respectively; (b) a singlet with satellites at 2.06 ppm ($^3J_{HPt}$ = 31.1 Hz), which integrates for six H nuclei, consistent with a Pt-bonded dimethylsulfide molecule; (c) a doublet resonance with satellites at 8.22 ppm ($^4J_{HP}$ = 2.3 Hz, $^3J_{HPt}$ = 78.6 Hz) that can be assigned to a proton at the *meta* position of an *ortho*-cyclometalated Ph$^{NO_2}$ ring. Interestingly, in agreement with the interpretation of the NMR data for

**PAr$^{Xyl_2}$(OPh$^{NO_2}$)$_2$**, the four methyl groups of the terphenyl ring in **1·SMe$_2$** give rise to two different signals in the corresponding NMR spectra, with a decrease in the apparent symmetry of the Ar$^{Xyl_2}$ moiety (from $C_{2v}$ to $C_2$) due to the metalation of one of the two Ph$^{NO_2}$ groups. This observation is consistent with the fast rotation around the P—C bond on the NMR time scale, as in the free ligand.

To confirm that **1·SMe$_2$** formed with the concomitant elimination of methane, the synthesis was performed in CD$_2$Cl$_2$ in a sealed NMR tube. Indeed, a singlet resonance at 0.21 ppm was detected in the $^1$H NMR spectrum of the reaction mixture after a few hours and assigned to CH$_4$ in dichloromethane solution according to the literature [34].

The synthesis of the carbonyl complex **1·CO** was straightforward by exposing dichloromethane solutions of **1·SMe$_2$** to 1 atm of CO at room temperature. Although **1·CO** could be isolated as a pure solid material by the fast elimination of the volatiles from the reaction mixture under reduced pressure, the appearance of a dark solid material indicated that some decomposition took place after 20–30 min under vacuum. The IR spectrum of **1·CO** in CH$_2$Cl$_2$ solution shows a strong absorption at 2077 cm$^{-1}$, which is in the expected range for Pt(II) carbonyl complexes. X-ray diffraction analyses permitted ascertaining the molecular structure of **1·CO** in the solid state (Figure 3). The platinum atom lies in a distorted square planar environment with the phosphonite ligand adopting a B conformation (C-C-P-Pt dihedral angle of ca. 40°). The smallest bond angle at the metal center, 80.86(6)°, corresponds to the five-member ring originated by the cyclometallation of one of the two Ph$^{NO_2}$ rings, whereas the largest one (ca. 100°) is that comprised by the CO ligand and the P atom and is likely influenced by the steric hindrance of the phosphorated ligand. The trend in the three different Pt-C bond distances parallels the hybridization of the C atom involved in the bond, i.e., Pt-C$sp$ < Pt-C$sp^2$ < Pt-C$sp^3$.

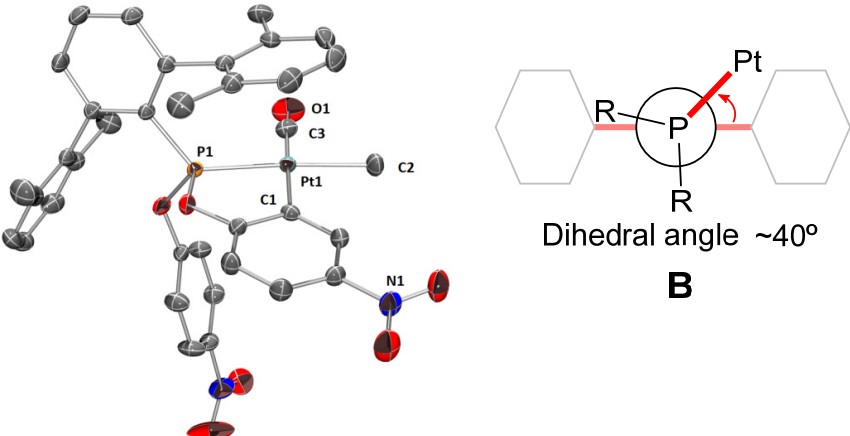

**Figure 3.** ORTEP view of the X-ray molecular structure of complex **1·CO**, with ellipsoids at the 50% probability level and H atoms omitted for clarity (**left**); schematic view of the B conformation adopted by the ligand **PAr$^{Xyl_2}$(OPh$^{NO_2}$)$_2$** (**right**) [18,26]. Selected bond distances (Å) and angles (°): Pt1-P1 2.2643(8), Pt1-C1 2.054(2), Pt1-C2 2.110(3), Pt1-C3 1.901(2), C3-O1 1.126(3); P1-Pt1-C1 80.86(6); C1-Pt1-C2 90.78(9); C2-Pt1-C3 88.06(1); C3-Pt1-P1 100.41(8).

Our previous studies on terphenyl phosphine complexes of platinum proved that, in some cases, these ligands can be forced to adopt a $\kappa^2$-P,C coordination mode in which one of the C atoms of a xylyl group is involved in a secondary interaction with the metal center [18–22]. So, we envisaged that a complex of this type could be prepared by reacting PtCl$_2$ with the phosphonites reported here. Accordingly, **PAr$^{Xyl_2}$(OPh$^{NO_2}$)$_2$** was treated with equimolar amounts of PtCl$_2$ in toluene at 100 °C (higher reaction temperatures are required because PtCl$_2$ is an insoluble polymeric material). Nevertheless, after 72 h, the ligand was quantitatively converted into the dinuclear platinum complex, **2**, which again resulted from the *ortho*-metalation of a Ph$^{NO_2}$ ring and elimination of HCl. Its characterization by NMR spectroscopy does not deserve any specific comment other than the dramatic

increase in the $^1J_{PPt}$ coupling constant with respect to complexes **1** (6222 Hz for **2**; 2550 and 2193 Hz for **1·SMe₂** and **1·CO**, respectively), as a consequence of the low trans influence of the bridging chloride ligands compared to the strong σ-donor methyl ligand [35].

The molecular structure of complex **2** was determined by X-ray diffraction studies and is represented in Figure 4 (left) together with schematic views of the coordination planes and the angle they form and with the binary axis $C_2$ (top right) and the conformation adopted by the phosphonite ligand (bottom right).

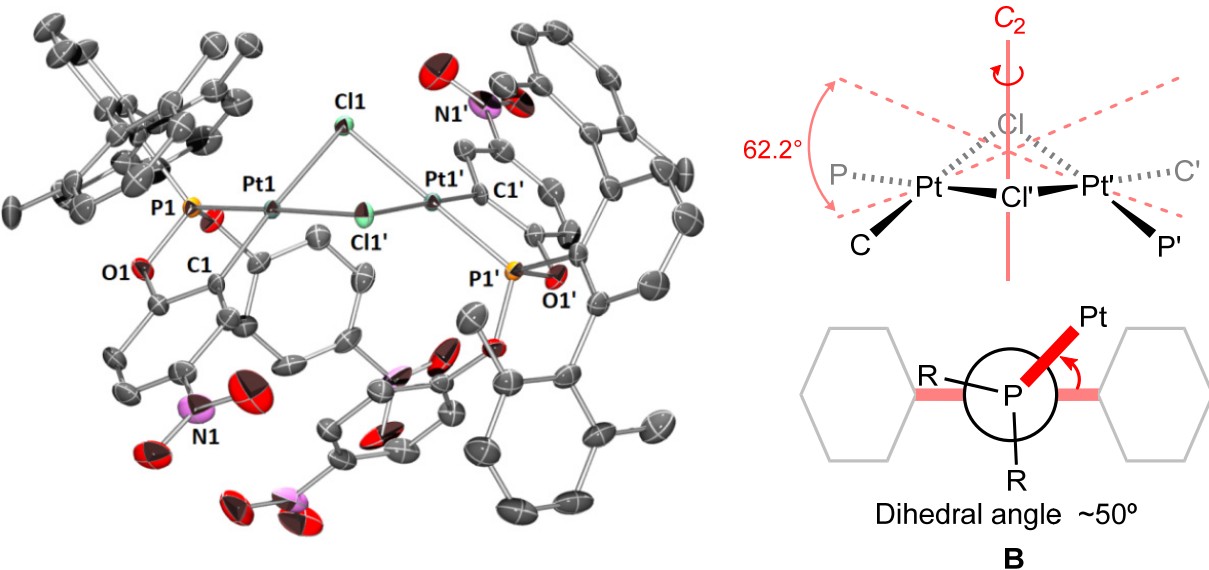

**Figure 4.** ORTEP view of the X-ray molecular structure of complex **2**, with ellipsoids at the 50% probability level and H atoms omitted for clarity (**left**); schematic view of the coordination planes in **2** with the angle they form and the binary axis $C_2$ (top **right**); schematic view of the B conformation adopted by the ligand **PAr^Xyl₂(OPh^NO₂)₂** [18,26]. Selected bond distances (Å) and angles (°): Pt1-P1 2.146(1), Pt1-C1 1.974(4), Pt1-Cl1 2.410(1), Pt1-Cl1′ 2.398(1); P1-Pt1-C1 82.6(1); C1-Pt1-Cl1′ 95.4(1); Cl-Pt1-Cl′ 82.89(4); Cl1-Pt1-P1 99.10(4); Pt1-Cl1-Pt1′ 82.60(4).

The asymmetric unit contains half a molecule of **2**, the second one being generated by a $C_2$ axis that forms an angle of ca. 30° with the coordination plane around the metal center, giving rise to a chiral species. The two coordination planes form an angle of ca. 60° (Figure 4, top right). Similarly to complex **1·CO**, the metalated phosphonite ligands adopt a B conformation with a C-C-P-Pt dihedral angle of ca. 50°. Both the Pt-P and Pt-C bond distances are significantly shorter in **2** than in the carbonyl compound, in accordance with the lower trans influence of the bridging chlorides compared to Me and CO.

The dinuclear complex **2** was easily converted into the mononuclear carbonyl chloride complex **2·CO** by treatment with CO at room temperature (Scheme 2). The configuration proposed in Scheme 2 for **2·CO**, i.e., a trans distribution of the P donor and CO ligand around the platinum center, is based on the higher values of the CO stretching vibration of **2·CO** compared to **1·CO** (2110 vs. 2077 cm$^{-1}$, respectively), in accordance with the presence of a π-acidic, poorly σ-donating phosphonite trans to the carbonyl ligand in **2·CO**. Moreover, the $^1J_{PPt}$ coupling of 4758 Hz is similar to that of the carbonyl complex **3·CO** (5023 Hz, vide infra), in which the P(III) and the CO ligands occupy mutually trans positions, and much lower than that of the chloride complex **2** (6222 Hz), in which the P donor is trans to a chloride ligand.

As mentioned in the Introduction, the main goal of our work on new bulky phosphonite ligands was the stabilization of homoleptic two-coordinate Pt(0) complexes. However, the reactions reported in Scheme 2 demonstrate that **PAr^Xyl₂(OPh^NO₂)₂** easily undergoes *ortho*-cyclometallation, which makes difficult any further reduction of the resulting complex. The use of the ligand **PAr^Xyl₂(OPh^NO₂,Me)₂**, with the four *ortho* positions of the

4-nitrophenoxy rings occupied by methyl groups, was expected to prevent cyclometalla­tion reactions and to provide more steric protection to the metal center. So, we treated the same starting materials shown Scheme 2, i.e., PtCl₂ and [PtMe₂(μ-SMe₂)]₂, with **PAr$^{Xyl_2}$(OPh$^{NO_2,Me}$)₂** at room temperature, but no reaction took place, likely as a con­sequence of the combined effects of low basicity and high steric demand of the phosphonite. While no coordination to platinum was detected using PtCl₂ as the starting material, even at higher temperatures (up to 100 °C), with [PtMe₂(μ-SMe₂)]₂, a new complex formed when heating 1:2 mixtures of [PtMe₂(μ-SMe₂)]₂ and **PAr$^{Xyl_2}$(OPh$^{NO_2,Me}$)₂** in THF at 60 °C, as revealed by the presence of a singlet at 165.6 ppm (¹*J*$_{PPt}$ of 6185 Hz) in the ³¹P{¹H} NMR spectrum of the crude. The slow conversion of the starting materials was completed after ca. 48 h. The new substance was isolated as a colorless, air-stable solid, characterized by NMR spectroscopy and X-ray diffraction analyses, and identified as the bis-cyclometalated Pt(II) complex, **3·SMe₂** (Scheme 3), which is the result of C-H activation of two methyl groups of the OPh$^{NO_2,Me}$ rings, elimination of methane, and eventual formation of two Pt-C*sp*³ bonds in a trans disposition.

**Scheme 3.** Synthesis of complexes **3**.

The most relevant NMR data for **3·SMe₂** are listed below:

- The four methyl groups of the xylyl rings give rise to one singlet at 2.10 ppm and one singlet at 21.5 ppm in the ¹H and ¹³C{¹H} NMR spectra, respectively, in accor­dance with fast rotation around the P—C bond on the NMR time-scale, as in the free phosphonite ligands;
- The two metalated OPh$^{NO_2,Me}$ fragments are equivalent, as shown by the presence of only one resonance for the CH₃ groups in both the ¹H- and ¹³C{¹H} NMR spectra;
- In the ¹H NMR spectrum, the four diastereotopic methylene protons are coupled with the ³¹P nucleus, thus constituting first-order AMX spin systems, which give rise to a doublet (²*J*$_{HH}$ = 11.5 Hz, ³*J*$_{HP}$ ≈ 0 Hz) and a triplet (²*J*$_{HH}$ ≈ ³*J*$_{HP}$ = 11.5 Hz), both with ¹⁹⁵Pt satellites;
- The six protons of the coordinated dimethylsulfide resonate as a doublet at 2.34 ppm with ⁴*J*$_{HP}$ = 4.0 Hz and ³*J*$_{HPt}$ = 36.4 Hz.

The molecular structure of **3·SMe₂** is represented in Figure 5 together with a schematic view of the conformation adopted by the phosphonite ligand, again of the B type with a C-C-P-Pt dihedral angle of approximately 40°.

The coordination geometry is best described as slightly distorted square planar, with the metal center lying at a distance of ca. 0.13 Å above the midplane defined by the four donor atoms. The two Pt—C bond distances are significantly different (ca. 0.03 Å), likely as a consequence of the different conformations adopted by the two six-membered rings to which they pertain. Such conformations are, in turn, influenced by the dihedral angle shown in Figure 6, which forces one of the two P—O bonds to a quasi-eclipsed conformation with the Pt—C2 bond (see Figure 6).

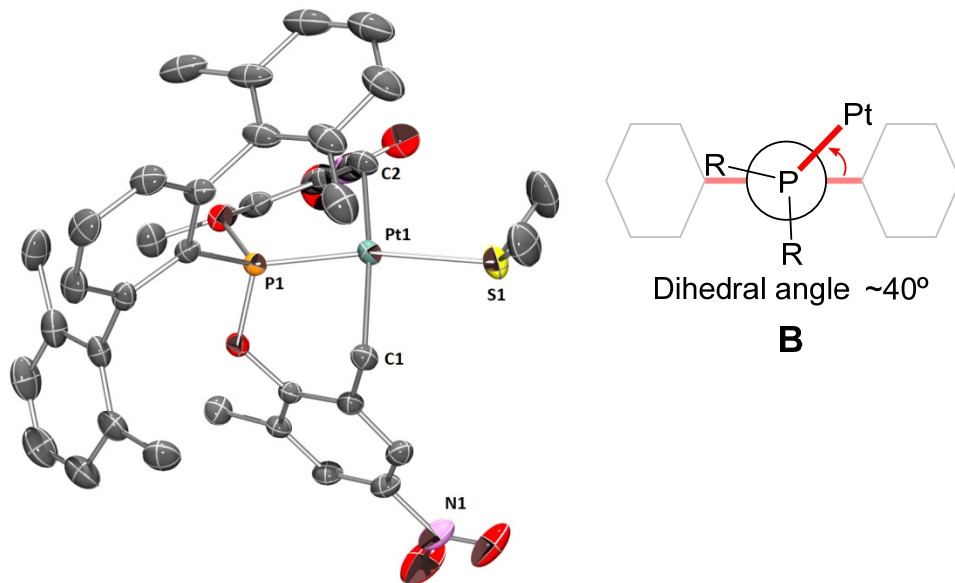

**Figure 5.** ORTEP view of the X-ray molecular structure of complex **3·SMe₂**, with ellipsoids at the 50% probability level and H atoms omitted for clarity (**left**); schematic view of the B conformation adopted by the ligand **PAr^Xyl₂(OPh^NO₂,Me)₂** (**right**) [17,25]. Selected bond distances (Å) and angles (°): Pt1-P1 2.1526(9), Pt1-C1 2.155(3), Pt1-C2 2.120(3), Pt1-S1 2.339(1); P1-Pt1-C1 84.18(7); C1-Pt1-S1 86.74(8); S1-Pt1-C2 99.09(9); C2-Pt1-P1 89.23(8); C1-Pt1-C2 169(1).

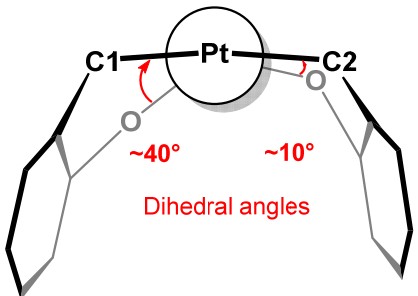

**Figure 6.** Schematic view along the Pt—P bond of the conformations adopted by the two cyclometalated 6-membered rings together with the values of the O-P-Pt-C dihedral angles.

The carbonyl complex **3·CO** was selectively obtained by exposing dichloromethane solutions of the parent dimethylsulfide adduct to 1 atm of CO (Scheme 3). The unexpectedly high value of 2140 cm⁻¹ found for the C≡O stretching vibration in the IR spectrum of **3·CO** in CH₂Cl₂ substantially matches that of free CO (2143 cm⁻¹), suggesting an almost negligible contribution of π-backdonation to the Pt—CO bonding, and exceeds by 70 cm⁻¹ that found for **1·CO**. This strong difference observed for two apparently similar platinum(II) carbonyls can reasonably be ascribed to the different electronic properties of the ligand trans to CO in each complex: a strongly σ-donating aryl ligand in **1·CO** and a P-bonded electron-poor phosphonite in **3·CO**, albeit in a dianionic form. In contrast to what was observed for **3·SMe₂**, the ¹H NMR spectrum of **3·CO** shows two well-separated doublets of doublets with ¹⁹⁵Pt satellites for the AMX spin system of the two equivalent methylene groups at 2.55 and 1.48 ppm (Figure 7).

The dimethylsulfide ligand was also easily displaced from **3·SMe₂** by triphenylphosphine, affording the air-stable complex **3·PPh₃**, whose ³¹P{¹H} NMR spectrum consists of two doublets at 180.8 (phosphonite) and 24.8 ppm (phosphane) with a large ²J_PP coupling constant of 528 Hz, in agreement with the mutual trans disposition of the two ³¹P nuclei. A significantly larger ³¹P-¹⁹⁵Pt coupling is observed for the phosphonite (4875 Hz) than for the phosphine (3101 Hz) [36]. Regarding the ¹H NMR of **3·PPh₃**, the two equivalent Pt-bonded methylene groups give rise to a doublet of doublets and a triplet of doublets,

centered at 1.63 and 1.47 ppm, respectively, which can be described together as a first-order AMPX spin system, once the $^{31}$P-$^{1}$H couplings have been considered. The two CH$_2$ protons show a geminal coupling of 10.8 Hz ($J_{AM}$), each of them being further coupled to the two inequivalent $^{31}$P nuclei with different $^{3}J_{HP}$ coupling constants ($J_{AP}$ and $J_{AX}$; $J_{MP}$ and $J_{MX}$). For the low-field signal (dd), the value of one of the two $^{3}J_{HP}$ is negligible, whereas for the up-field one (td), the geminal $^{1}$H-$^{1}$H ($J_{AM}$) and one of the two $^{1}$H-$^{31}$P coupling constants have very similar values, thus originating a signal resembling a triplet of doublets.

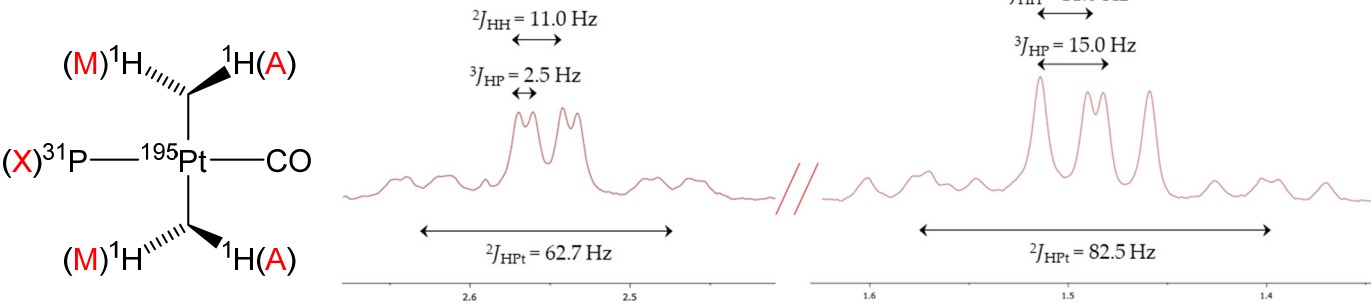

**Figure 7.** $^{1}$H NMR signals of the AMX spin system due to the four Pt-CH$_2$ protons in **3·CO** (CD$_2$Cl$_2$) with the corresponding $^{1}$H-$^{1}$H (A-M), $^{1}$H-$^{31}$P (A-X and M-X), and $^{1}$H-$^{195}$Pt couplings.

## 3. Materials and Methods

All manipulations were carried out using standard Schlenk and glove-box techniques, under an atmosphere of high-purity nitrogen. Solvents were rigorously dried and degassed before use. Compounds [PtMe$_2$($\mu$-SMe$_2$)]$_2$ [37] and PX$_2$Ar$^{Xyl_2}$ (a mixture of PCl$_2$Ar$^{Xyl_2}$, PBr$_2$Ar$^{Xyl_2}$, and PClBrAr$^{Xyl_2}$) [22] were prepared according to previously reported procedures. Other chemicals were commercially available and used as received. $^{1}$H-, $^{13}$C-, and $^{31}$P NMR spectra were recorded at 300 and 400 MHz, using the solvent peak as the internal reference. Spectral assignments were made by routine one- and two-dimensional NMR experiments ($^{1}$H, $^{13}$C{$^{1}$H}, $^{31}$P{$^{1}$H}, COSY, HSQC and HMBC) where appropriate. All $^{3}J_{HH}$ aromatic-coupling constants in $^{1}$H NMR spectra are ca. 7–8 Hz. Ph$^{NO2'}$ and Ph$^{NO_2,Me'}$ indicate the metalated Ph$^{NO_2}$ and Ph$^{NO_2,Me}$ groups, respectively. For elemental analyses, a LECO TruSpec CHN elementary analyzer was utilized.

The geometries of the phosphorus ligands were optimized in the gas phase without geometric restrictions using DFT with the Gaussian 09 program [38]. The functional used for these calculations is $\omega$B97XD [39] with the set of basis functions 6-31g(d,p) for C, H, O, N, and P atoms [40]. %Vbur data were obtained with the web application SambVca 2.1 [41,42]. The %G descriptor gives the area of the shadow cast by the ligand onto a sphere with transition metal or a focal point at the origin and was calculated with the Solid-G program [43,44]. To make the results comparable with the literature, a focal point was added to the geometries optimized by DFT next to the phosphorus atoms at a distance of 2.28 Å, which was used for the calculation of the TCA, %Vbur, and %G parameters [45].

A summary of the crystallographic data and the structure refinement results for compounds **1·CO**, **2** and **3·SMe$_2$** is given in Tables S1–S3 (ESI). Crystals of suitable size for X ray diffraction analysis were coated with dry perfluoropolyether and mounted on glass fibers and fixed in a cold nitrogen stream (T = 213 K) to the goniometer head. Data collection was carried out on a Bruker-AXS, D8 QUEST ECO, PHOTON II area detector diffractometer, using monochromatic radiation $\lambda$(Mo K$\alpha$) = 0.71073 Å, by means of $\omega$ and $\varphi$ scans with a width of 0.50 degrees. The data were reduced (SAINT) [46] and corrected for absorption effects by the multi-scan method (SADABS) [46,47]. The structures were solved by direct methods (SIR-2002) [48] and refined against all F$^2$ data by full-matrix least-squares techniques (SHELXTL-2018/3) [49,50] minimizing w[Fo$^2$-Fc$^2$]$^2$. All non-hydrogen atoms were refined anisotropically. Hydrogen atoms were included from calculated positions and refined riding on their respective carbon atoms with isotropic

displacement parameters. CCDC 2182904 (**1·CO**), 2182905 (**2**), and 2182906 (**3·SMe$_2$**) contain the supplementary crystallographic data for this paper. The data can be obtained free of charge via: https://www.ccdc.cam.ac.uk/structures/ (Accessed on 29 July 2022).

**Synthesis of PAr$^{Xyl2}$(OPh$^{NO_2}$)$_2$.** PX$_2$Ar$^{Xyl2}$ (1.5 mmol, 580 mg), 4-nitrophenol (3.0 mmol, 420 mg), and triethylamine (4.0 mmol, 0.50 mL) were dissolved in diethylether (10 mL) and stirred at room temperature for ca. 12 h. Solid materials were filtered off and all volatiles were removed under reduced pressure to give **PAr$^{Xyl2}$(OPh$^{NO_2}$)$_2$**, as a yellow crystalline solid (580 mg, 65%). $^1$H NMR (400 MHz, CDCl$_3$): $\delta$ 7.92 (d, 4H, $m$-Ph$^{NO_2}$), 7.65 (t, 1H, $p$-C$_6$H$_3$), 7.21 (t, 2H, $p$-Xyl), 7.18 (dd, 2H, $^3J_{HP}$ = 2.8 Hz, $m$-C$_6$H$_3$) 7.08 (d, 4H, $m$-Xyl), 6.45 (d, 4H, $o$-Ph$^{NO_2}$), 2.08 (s, 12H, CH$_3$) ppm. $^{31}$P{$^1$H} NMR (162 MHz, CDCl$_3$): $\delta$ 165.7 (s) ppm. $^{13}$C{$^1$H} NMR (101.0 MHz, CDCl$_3$): $\delta$ 160.0 (d, $^2J_{CP}$ = 10.3 Hz, C-O), 145.9 (d, $^2J_{CP}$ = 23 Hz, $o$-C$_6$H$_3$), 143.4 (s, C-N), 140.7 (d, $^3J_{CP}$ = 6 Hz, ipso-Xyl), 136.5 (s, $o$-Xyl), 132.9 (d, $^1J_{CP}$ = 15 Hz, $C$-$P$), 130.0 (d, $^3J_{CP}$ = 3 Hz, $m$-C$_6$H$_3$), 127.8 (s, $p$-Xyl), 127.3 (s, $m$-Xyl), 125.6 (s, $m$-Ph$^{NO_2}$), 118.9 (d, $^3J_{CP}$ = 11 Hz, $o$-Ph$^{NO_2}$), 21.2 (s, CH$_3$) ppm. Anal. Calc. for C$_{34}$H$_{29}$N$_2$O$_6$P: C, 68.91; H, 4.93; N, 4.73. Found: C, 68.9; H, 4.9; N, 4.6.

**Synthesis of PAr$^{Xyl2}$(OPh$^{NO_2,Me}$)$_2$.** PX$_2$Ar$^{Xyl}$ (1.3 mmol, 500 mg), 2,6-dimethyl-4-nitrophenol (2.6 mmol, 440 mg), triethylamine (4.0 mmol, 0.50 mL) were dissolved in diethylether (10 mL) and stirred at 70 °C for ca. 48 h ($^{31}$P NMR spectrum of the reaction mixture was recorded to confirm full conversion of PX$_2$Ar$^{Xyl2}$). Solid materials were filtered off and all volatiles were removed under reduced pressure to give a sticky yellow solid, which was washed with pentane (2 × 3 mL) and dried under vacuum to afford pure **PAr$^{Xyl2}$(OPh$^{NO_2,Me}$)$_2$** as a yellow crystalline solid (590 mg, 70%). $^1$H NMR (400 MHz, CD$_2$Cl$_2$): $\delta$ 7.73 (t, 1H, $p$-C$_6$H$_3$), 7.64 (s, 4H, $m$-Ph$^{NO_2,Me}$), 7.20 (dd, 2H, $^3J_{HP}$ = 2.7 Hz, $m$-C$_6$H$_3$), 7.10 (t, 2H, $p$-Xyl), 7.00 (d, 4H, $m$-Xyl), 2.14 (s, 12H, CH$_3$ Xyl), 1.86 (s, 6H, CH$_3$ Ph$^{NO_2,Me}$), 1.85 (s, 6H, CH$_3$ Ph$^{NO_2,Me}$) ppm. $^{31}$P{$^1$H} NMR (162 MHz, CDCl$_3$) $\delta$ 174.2 (s) ppm. $^{13}$C{$^1$H} NMR (101 MHz, CDCl$_3$) $\delta$ 156.2 (d, $^2J_{CP}$ = 5 Hz, C-O), 145.9 (d, $^2J_{CP}$ = 23 Hz, $o$-C$_6$H$_3$), 142.9 (s, C-N), 141.1 (d, $^3J_{CP}$ = 5 Hz, ipso-Xyl), 136.7 (s, $o$-Xyl), 135.2 (d, $^1J_{CP}$ = 29 Hz, C-P), 132.5 (d, $^4J_{CP}$ = 2 Hz, $p$-C$_6$H$_3$), 130.3 (br s, $m$-C$_6$H$_3$), 127.9 (s, $p$-Xyl), 127.4 (s, $m$-Xyl), 124.7 (s, $o$-Ph$^{NO_2,Me}$), 21.6 (s, CH$_3$ Xyl), 19.0 (s, CH$_3$ Ph$^{NO_2,Me}$), 18.9 (s, CH$_3$ Ph$^{NO_2,Me}$) ppm. Anal. Calc. for C$_{38}$H$_{37}$N$_2$O$_6$P: C, 70.36; H, 5.75; N, 4.32. Found: C, 70.2; H, 5.7; N, 4.4.

**Synthesis of complex 1·SMe$_2$.** Solid samples of **PAr$^{Xyl2}$(OPh$^{NO_2}$)$_2$** (0.50 mmol, 300 mg) and [PtMe$_2$(μ-SMe$_2$)]$_2$ (0.25 mmol, 140 mg) were dissolved in CH$_2$Cl$_2$ (10 mL) at room temperature. After 12 h stirring, all the volatiles were removed under reduced pressure to afford a colorless solid, which was washed with pentane (2 × 5 mL) and dried under vacuum. Crystallization from pentane/CH$_2$Cl$_2$ mixtures under nitrogen at −20 °C provided analytically pure samples of **1·SMe$_2$** (65%, 280 mg). $^1$H NMR (400 MHz, CD$_2$Cl$_2$): $\delta$ 8.22 (d, 1H, $^4J_{HP}$ = 2.3 Hz, $^3J_{HPt}$ = 78.6 Hz, $m$-Ph$^{NO_{2'}}$), 7.92-7.80 (m, 3H, $m$-Ph$^{NO_2}$ and $m$-Ph$^{NO_{2'}}$), 7.68 (t, 1H, $p$-C$_6$H$_3$), 7.17 (dd, 2H, $^4J_{HP}$ = 3.8 Hz, $m$-C$_6$H$_3$), 7.01-6.88 (m, 5H, $m$-Xyl and $o$-Ph$^{NO_{2'}}$), 6.75 (br t, 2H, $p$-Xyl), 6.53 (d, 2H, $o$-Ph$^{NO_2}$), 2.19 (s, 6H, CH$_3$ Xyl), 2.06 (s, 6H, $^3J_{HPt}$ = 31.1 Hz, SMe$_2$), 1.98 (s, 6H, CH$_3$ Xyl), 0.43 (d, $^3J_{HP}$ = 8.6 Hz, $^2J_{HPt}$ = 65.1 Hz, 3H, Pt-CH$_3$) ppm. $^{31}$P{$^1$H} NMR (162 MHz, CD$_2$Cl$_2$): $\delta$ 169.3 (s, $^1J_{PPt}$ = 2550 Hz) ppm. $^{13}$C{$^1$H} NMR (101 MHz, CD$_2$Cl$_2$): $\delta$ 167.6 (d, $^2J_{CP}$ = 21 Hz, C-Pt), 157.4 (d, $^2J_{CP}$ = 3 Hz, C-O), 146.3 (d, $^2J_{CP}$ = 15 Hz, $o$-C$_6$H$_3$), 144.1 (s, C-N) 142.7 (s, C-N), 141.8 (d, $^3J_{CP}$ = 5 Hz, ipso-Xyl), 136.4 (s, $o$-Xyl), 136.0 (s, $o$-Xyl), 133.1 (d, $^4J_{CP}$ = 2 Hz, $p$-C$_6$H$_3$), 131.6 (d, $^3J_{CP}$ = 9 Hz, $m$-C$_6$H$_3$), 128.7 (d, $^3J_{CP}$ = 2 Hz, $^2J_{CPt}$ = 46 Hz, $m$-Ph$^{NO_{2'}}$), 127.7 (s, $m$-Xyl), 127.6 (s, $m$-Xyl), 127.5 (s, $p$-Xyl), 125.2 (s, $m$-Ph$^{NO_2}$), 121.8 (d, $^3J_{CP}$ = 8 Hz, $o$-Ph$^{NO_2}$), 112.1 (d, $^3J_{CP}$ = 13 Hz, $o$-Ph$^{NO_{2'}}$), 22.5 (br s, SMe$_2$), 22.0 (s, CH$_3$ Xyl), 21.4 (s, CH$_3$ Xyl), 8.9 (d, $^2J_{CP}$ = 117 Hz, $^1J_{CPt}$ = 640 Hz, Pt-CH$_3$) ppm. ipso-C$_6$H$_3$ and one of the C-O were not detected or masked by more intense signals. Anal. Calc. for C$_{37}$H$_{37}$N$_2$O$_6$PPtS: C, 51.45; H, 4.32; S, 3.71; N, 3.24. Found: C, 51.6; H, 4.5; S, 3.4; N, 3.2.

**Synthesis of complex 1·CO**. A solution of **1·SMe$_2$** (60 mg, 0.070 mmol) in CH$_2$Cl$_2$ (3 mL) was degassed under reduced pressure and exposed to CO (1 atm) for ca. 12 h. The volatiles were quickly removed under reduced pressure to yield a light-grey solid material, which was identified as **1·CO** (50 mg, 86%). Single crystals suitable for X-ray diffraction

were grown at −20 °C by the gas-phase diffusion of pentane into a $CH_2Cl_2$ solution of the compound. $^1$H NMR (400 MHz, $CD_2Cl_2$): $\delta$ 8.26 (d, 1H, $^4J_{HP}$ = 2.6 Hz, $^3J_{HPt}$ = 66.4 Hz, *m*-Ph$^{NO_2'}$), 8.05–7.92 (m, 3H, *m*-Ph$^{NO_2'}$ + *m*-Ph$^{NO_2}$), 7.78 (td, $^5J_{HP}$ = 0.9 Hz, *p*-$C_6H_3$), 7.24 (dd, 2H, $^4J_{HP}$ = 4.2 Hz, *m*-$C_6H_3$), 7.13 (br d, 4H, *m*-Xyl), 7.00 (d, 1H, *o*-Ph$^{NO_2'}$), 6.85 (br t, 2H, *p*-Xyl), 6.51 (d, 2H, *o*-Ph$^{NO_2}$), 2.19 (s, 6H, $CH_3$ Xyl), 1.89 (s, 6H, $CH_3$ Xyl), 0.59 (d, $^3J_{HP}$ = 7.9 Hz, $^2J_{HPt}$ = 68.4 Hz, Pt-$CH_3$) ppm. $^{31}$P{$^1$H} NMR (162 MHz, $CD_2Cl_2$): $\delta$ 173.8 (s, $^1J_{PPt}$ = 2193 Hz) ppm. $^{13}$C{$^1$H} NMR (101 MHz, $CD_2Cl_2$): $\delta$ 169.6 (d, $^2J_{CP}$ = 21 Hz, C-Pt), 156.1 (d, $^2J_{CP}$ = 16 Hz, C-O), 146.5 (d, $^2J_{CP}$ = 13 Hz, *o*-$C_6H_3$), 145.2 (s, C-N), 143.0 (s, C-N), 141.0 (d, $^3J_{CP}$ = 4 Hz, ipso-Xyl), 136.5 (s, *o*-Xyl), 136.3 (s, *o*-Xyl), 134.4 (s, *p*-$C_6H_3$), 131.4 (d, $^3J_{CP}$ = 9 Hz, *m*-$C_6H_3$), 128.8 (s, $^2J_{CPt}$ = 48 Hz, *m*-Ph$^{NO_2'}$), 128.1 (s, *m*-Xyl), 127.9 (s, *m*-Xyl), 127.7 (s, *p*-Xyl), 125.3 (s, *m*-Ph$^{NO_2}$), 124.3 (s, *m*-Ph$^{NO_2'}$), 121.4 (d, $^3J_{CP}$ = 4 Hz, *o*-Ph$^{NO_2}$), 112.0 (d, $^3J_{CP}$ = 13 Hz, *o*-Ph$^{NO_2'}$), 21.9 (s, $CH_3$ Xyl), 21.2 (s, $CH_3$ Xyl), 0.4 (d, $^2J_{CP}$ = 98 Hz, $^1J_{CPt}$ = 533 Hz, Pt-$CH_3$) ppm. Pt-CO, one C-O, and ipso-$C_6H_3$ not detected or masked by more intense signals. IR ($CH_2Cl_2$): 2077 cm$^{-1}$ ($\tilde{\nu}_{C\equiv O}$).

**Synthesis of complex 2.** A solution of **PAr$^{Xyl_2}$(OPh$^{NO_2}$)$_2$** (0.50 mmol, 300 mg) in 10 mL of toluene was added to solid $PtCl_2$ (0.50 mmol, 130 mg) and the resulting suspension was stirred for 72 h at 100 °C. The mixture was allowed to reach room temperature and the volatiles were removed under reduced pressure. The resulting white solid was washed with pentane (2 × 5 mL). Crystallization from pentane/$CH_2Cl_2$ mixtures under nitrogen at −20 °C provided analytically pure samples of the the dinuclear complex **2** (670 mg, 82%). $^1$H NMR (400 MHz, CDCl$_3$): $\delta$ 7.96 (d, 2H, $^4J_{HP}$ = 6.7 Hz, $^3J_{HPt}$ = 144.0 Hz, *m*-Ph$^{NO_2'}$), 7.81–7.70 (m, 6H, *o*-Ph$^{NO_2'}$ + *m*-Ph$^{NO_2}$), 7.27 (dd, 4H, $^4J_{HP}$ = 4.2 Hz, *m*-$C_6H_3$), 6.98 (t, 4H, *p*-Xyl), 6.90 (t, 2H, *p*-$C_6H_3$), 6.71 (br d, 8H, *m*-Xyl), 6.37 (d, 4H, *o*-Ph$^{NO_2}$), 2.23 (s, 12H, $CH_3$ Xyl), 1.93 (s, 12H, $CH_3$ Xyl) ppm. $^{31}$P{$^1$H} NMR (162 MHz, CDCl$_3$): $\delta$ 121.7 (s, $^1J_{PPt}$ = 6222 Hz) ppm. $^{13}$C{$^1$H} NMR (101 MHz, CDCl$_3$): $\delta$ 164.5 (d, $^2J_{CP}$ = 11 Hz, $^1J_{CPt}$ = 59 Hz C-Pt), 154.5 (s, C-N), 146.9 (d, $^2J_{CP}$ = 17 Hz, *o*-$C_6H_3$), 144.2 (s, *o*-Xyl), 142.3 (s, *m*-Ph$^{NO_2'}$), 140.8 (d, $^3J_{CP}$ = 3 Hz, ipso-Xyl), 136.1 (s, *p*-$C_6H_3$), 134.6 (s, *m*-Ph$^{NO_2}$), 131.3 (d, $^3J_{CP}$ = 11 Hz, *m*-$C_3H_6$), 127.9 (*p*-Xyl), 127.4 (d, $^1J_{CP}$ = 32 Hz, ipso-$C_3H_6$), 126.3 (s, *o*-Ph$^{NO_2'}$), 125.4 (d, $^2J_{CP}$ = 10 Hz, C-O Ph$^{NO_2'}$), 124.6 (s, *m*-Xyl), 122.4 (br s, *m*-Ph$^{NO_2'}$), 121.75 (s, *o*-Ph$^{NO_2}$), 110.8 (d, $^3J_{CP}$ = 16 Hz, *o*-$C_3H_6$), 22.1 (s, $CH_3$), 22.1 (s, $CH_3$) ppm. Anal. Calc. for $C_{68}H_{56}Cl_2N_4O_{12}P_2Pt_2$: C, 49.67; H, 3.43; N, 3.41. Found: C, 49.3; H, 3.6; N, 3.4.

**Synthesis of complex 2·CO.** Complex **2** (40 mg, 0.025 mmol) was dissolved in $CH_2Cl_2$ (2 mL) under CO (1 atm). After 5 h, the reaction mixture was taken to dryness under reduced pressure to afford a colorless solid (40 mg, 97%). $^1$H NMR (400 MHz, $CD_2Cl_2$): $\delta$ 8.01 (d, 2H, *m*-Xyl), 7.92 (t, 1H, *p*-$C_6H_3$), 7.37 (t, 2H, *p*-Xyl), 7.28–7.10 (m, 5H, *m*-Xyl + *m*-$C_6H_3$ + *o*-Ph$^{NO_2'}$), 7.06 (d, 1H, *m*-Ph$^{NO_2'}$), 6.89 (s, *m*-Ph$^{NO_2}$), 6.60 (d, 2H, *o*-Ph$^{NO_2}$), 2.25 (s, 6H, $CH_3$ Xyl), 1.96 (s, 6H, $CH_3$ Xyl) ppm. $^{31}$P{$^1$H} NMR (162 MHz, $CD_2Cl_2$): $\delta$ 121.7 (s, $^1J_{PPt}$ = 4758 Hz) ppm. $^{13}$C{$^1$H} NMR (101 MHz, $CD_2Cl_2$): $\delta$ 176.5 (d, $^2J_{CP}$ = 6 Hz, CO), 166.3 (d, $^2J_{CP}$ = 13 Hz, $^1J_{CPt}$ = 51 Hz C-Pt), 155.5 (s, C-N), 147.1 (d, $^2J_{CP}$ = 15 Hz, C-O Ph$^{NO_2}$), 145.9 (s, *o*-Xyl), 143.4 (s, $^3J_{CPt}$ = 48 Hz, *m*-Ph$^{NO_2'}$), 140.1 (d, $^3J_{CP}$ = 4 Hz, ipso-Xyl), 136.6 (s, *p*-Xyl), 136.4 (s, *p*-Xyl), 135.6 (s, *p*-$C_6H_3$), 132.0 (d, $^3J_{CP}$ = 12 Hz, *m*-$C_3H_6$), 130.2 (s, *o*-Ph$^{NO_2}$), 128.8 (s, *m*-Ph$^{NO_2}$), 128.5 (d, $^1J_{CP}$ = 81 Hz, ipso-$C_6H_3$), 128.3 (s, *m*-Xyl), 128.2 (s, *m*-Xyl), 126.2 (s, *o*-Xyl), 121.4 (d, $^3J_{CP}$ = 5 Hz, *o*-Ph$^{NO_2}$), 111.7 (d, $^3J_{CP}$ = 17 Hz, *o*-$C_3H_6$), 21.9 (s, $CH_3$ Xyl), 21.2 (s, $CH_3$ Xyl) ppm. C-O Ph$^{NO2'}$ and one of the two C-N not detected or masked by more intense signals. IR ($CH_2Cl_2$): 2110 cm$^{-1}$ ($\tilde{\nu}_{C\equiv O}$).

**Synthesis of complex 3·SMe$_2$.** Solid samples of **PAr$^{Xyl_2}$(OPh$^{NO_2,Me}$)$_2$** (0.50 mmol, 320 mg) and [PtMe$_2$(μ-SMe$_2$)]$_2$ (0.25 mmol, 140 mg) were dissolved in THF (10 mL) and stirred at 60 °C for ca. 48 h. All the volatiles were then removed under reduced pressure to afford a colorless solid, which was washed with pentane (2 × 5 mL) and dried under vacuum. Crystallization from pentane/$CH_2Cl_2$ mixtures under nitrogen at −20 °C provided analytically pure samples of **3·SMe$_2$** (320 mg, 70%). $^1$H NMR (400 MHz, CDCl$_3$): $\delta$ 7.71 (t, 1H, *p*-$C_6H_3$), 7.52 (br s, 2H, *m*-Ph$^{NO_2,Me'}$), 7.43 (br s, 2H, *m*-Ph$^{NO_2,Me'}$), 7.17 (m, 4H, *m*-$C_6H_3$ and *p*-Xyl), 7.04 (d, 4H, *m*-Xyl), 2.34 (d, 6H, $^4J_{HP}$ = 4.0 Hz, $^3J_{HPt}$ = 36.4 Hz, SMe$_2$), 2.10 (s, 12H, $CH_3$ Xyl), 1.73 (s, 6H, $CH_3$ Ph$^{NO_2,Me'}$), 1.60 and 1.43 (AMX spin system, 4H,

$^2J_{HH}$ = 11.5 Hz, $^3J_{HP} \approx$ 0 Hz and $^3J_{HP} \approx$ 11.5 Hz, $^2J_{HPt}$ = 74.2 and 79.0 Hz, Pt-CH$_2$) ppm. $^{31}$P{$^1$H} NMR (162 MHz, CDCl$_3$): $\delta$ 165.6 ($^1J_{PPt}$ = 6185 Hz) ppm. $^{13}$C{$^1$H} NMR (101 MHz, CDCl$_3$): $\delta$ 156.1 (s, C-O), 146.8 (s, C-N), 142.6 (d, $^3J_{CP}$ = 6 Hz, ipso-Xyl), 142.1 (d, $^2J_{CP}$ = 4 Hz, *o*-C$_6$H$_3$), 136.6 (s, *o*-Xyl), 133.1 (s, *p*-C$_6$H$_3$), 131.2 (s, *m*-Ph$^{NO_2,Me'}$), 131.1(s, *m*-Ph$^{NO_2,Me'}$), 128.9 (s, *m*-C$_6$H$_3$), 127.8 (s, *p*-Xyl), 127.7 (s, *m*-Xyl), 120.1 (s, *C*-CH$_2$), 119.9 (s, *o*-Ph$^{NO_2,Me'}$), 21.5 (s, CH$_3$ Xyl), 21.1 (s, CH$_3$ Ph$^{NO_2,Me'}$), 16.8 (d, $^2J_{CP}$ = 6 Hz, Pt-CH$_2$), 14.6 (s, SMe$_2$) ppm. Anal. Calc. for C$_{40}$H$_{41}$N$_2$O$_6$PPtS: C, 53.15; H, 4.57; N, 3.10. Found: C, 52.9; H, 4.9; N, 3.0.

**Synthesis of 3·CO.** A solution of **3·SMe$_2$** (60 mg, 0.066 mmol) in CH$_2$Cl$_2$ (3 mL) was degassed under reduced pressure and exposed to CO (1 atm) for 1 h. The volatiles were quickly removed under reduced pressure to yield a colorless solid material, which was identified as **3·CO** (50 mg, 86%). $^1$H NMR (400 MHz, CD$_2$Cl$_2$): $\delta$ 7.82 (td, 1H, $^5J_{HP}$ = 0.9 Hz, *p*-C$_6$H$_3$), 7.62 (d, 2H, $^4J_{HP}$ = 2.7 Hz, *m*-Ph$^{NO_2,Me'}$), 7.46 (d, 2H, $^4J_{PH}$ = 2.0 Hz, *m*-Ph$^{NO_2,Me'}$), 7.25 (dd, 2H, $^4J_{HP}$ = 4.5 Hz, *m*-C$_6$H$_3$), 7.22 (t, 2H, *p*-Xyl), 7.10 (d, 4H, *m*-Xyl), 2.55 and 1.48 (AMX spin system, 4H, $^2J_{HH}$ = 11.0 Hz, $^3J_{HP}$ = 2.5 Hz and $^3J_{HP}$ = 15.0 Hz, $^2J_{HPt}$ = 62.7 and 82.5 Hz, Pt-CH$_2$), 2.10 (s, 12H, CH$_3$ Xyl), 1.73 (s, 6H, CH$_3$ Ph$^{NO_2,Me'}$) ppm. $^{31}$P{$^1$H} NMR (162 MHz, CD$_2$Cl$_2$): $\delta$ 181.2 ($^1J_{PPt}$ = 5023 Hz) ppm. $^{13}$C{$^1$H} NMR (101 MHz, CD$_2$Cl$_2$): $\delta$ 155.9 (d, $^2J_{CP}$ = 5 Hz, $^1J_{CPt}$ = 43 Hz, C-O), 147.1 (d, $^2J_{CP}$ = 15 Hz, $^2J_{CPt}$ = 14 Hz, C-P), 145.9 (s, C-N), 143.6 (s, ipso-Xyl), 140.8 (d, $^2J_{CP}$ = 4 Hz, *o*-C$_6$H$_3$), 136.6 (d, $^4J_{CP}$ = 5 Hz, *o*-Xyl), 134.1 (s, *p*-C$_6$H$_3$), 131.2 (d, $^4J_{CP}$ = 10 Hz, *m*-Ph$^{NO_2,Me'}$), 128.3 (s, *p*-Xyl), 127.9 (s, *m*-Xyl), 127.0 (s, *m*-C$_6$H$_3$), 120.9 (s, $^2J_{CPt}$ = 20 Hz, *C*-CH$_2$), 120.7 (s, *o*-Ph$^{NO_2,Me'}$), 21.1 (s, CH$_3$ Xyl), 16.4 (s, CH$_3$ Ph$^{NO_2,Me'}$), 7.2 (d, $^2J_{CP}$ = 8 Hz, $^1J_{CPt}$ = 360 Hz, Pt-CH$_2$) ppm. IR (CH$_2$Cl$_2$): 2140 cm$^{-1}$ ($\widetilde{\nu}_{C\equiv O}$).

**Synthesis of 3·PPh$_3$.** Solid PPh$_3$ (80 mg, 0.30 mmol) was added to a solution of **3·SMe$_2$** (270 mg, 0.30 mmol) in dichloromethane (5 mL). After 1 h stirring at room temperature, the reaction mixture was taken to dryness under reduced pressure to yield a white solid, which was washed with pentane (2 × 5 mL) and dried under vacuum (230 mg, 70%). $^1$H NMR (400 MHz, CD$_2$Cl$_2$): $\delta$ 7.73 (t, 1H, *p*-C$_6$H$_3$), 7.39 (t, 3H, *p*-PPh$_3$), 7.34 (d, 2H, $^4J_{HH}$ = 2.7 Hz, *m*-Ph$^{NO_2,Me'}$), 7.27 (td, 6H, $^4J_{PH}$ = 1.4 Hz, *m*-PPh$_3$), 7.21 (t, 2H, *p*-Xyl), 7.18 (d, 2H, *m*-C$_6$H$_3$), 7.05 (d, 4H, *m*-Xyl), 6.92 (dd, 6H, $^3J_{PH}$ = 1.3 Hz, *o*-PPh$_3$), 6.10 (d, 2H, $^4J_{HH}$ = 2.7 Hz, *m*-Ph$^{NO_2,Me'}$), 2.16 (s, 12H, CH$_3$ Xyl), 1.74 (s, 6H, CH$_3$ Ph$^{NO_2,Me'}$), 1.63 and 1.47 (AMPX spin system, 4H, $^2J_{HH}$ = 10.8 Hz; $^3J_{HP}$ = 16.4 and 2.0 Hz, $^3J_{HP} \approx {}^2J_{HH}$ and $^3J_{HP}$ = 3.4 Hz; $^2J_{HPt}$ not measurable for overlapping and $^2J_{HPt}$ = 66.1 Hz, Pt-CH$_2$) ppm. $^{31}$P{$^1$H} NMR (162 MHz, CDCl$_3$): $\delta$ 180.8 (d, $^2J_{PP}$ = 528 Hz, $^1J_{PPt}$ = 4875 Hz, phosphonite), 24.8 (d, $^2J_{PP}$ = 528 Hz, $^1J_{PPt}$ = 3101 Hz, PPh$_3$) ppm. $^{13}$C{$^1$H} NMR (101 MHz, CDCl$_3$): $\delta$ 155.7 (d, $^2J_{CP}$ = 4 Hz, C-O), 147.1 (dd, $^1J_{CP}$ = 16 Hz, $^3J_{CP}$ = 1 Hz, ipso-C$_6$H$_3$), 142.7 (s, C-N), 142.6 (d, $J_{CP}$ = 7 Hz ipso-Xyl or *o*-C$_6$H$_3$), 141.8 (d, $J_{CP}$ = 5 Hz, ipso-Xyl or *o*-C$_6$H$_3$), 136.7 (s, *o*-Xyl), 134.2 (d, $^2J_{CP}$ = 12 Hz, *o*-PPh$_3$), 133.2 (s, *p*-C$_6$H$_3$), 131.2 (d, $^2J_{CP}$ = 10 Hz, *m*-C$_6$H$_3$), 130.7 (s, *p*-PPh$_3$), 129.2 (dd, $^1J_{CP}$ = 50 Hz, $^3J_{CP}$ = 3 Hz, ipso-PPh$_3$), 128.4 (s, *o*-Ph$^{NO_2,Me'}$), 128.4 (d, $^3J_{CP}$ = 4 Hz, *C*-CH$_2$), 128.3 (d, $^3J_{CP}$ = 10 Hz, *m*-PPh$_3$), 127.9 (s, *m*-Xyl), 127.7 (s, *p*-Xyl), 119.8 (s, *m*-Ph$^{NO_2,Me'}$), 119.3 (s, *m*-Ph$^{NO_2,Me'}$), 21.6 (s, CH$_3$ Xyl), 16.8 (s, CH$_3$ Ph$^{NO_2,Me'}$), 12.0 (t, $^2J_{CP}$ (PPh$_3$)~$^2J_{CP}$ (phosphonite) = 6 Hz, $^1J_{CPt}$ = 392 Hz, Pt-CH$_2$) ppm. Anal. Calc. for C$_{56}$H$_{55}$N$_2$O$_6$P$_2$Pt: C, 60.65; H, 5.00; N, 2.54 Found: C, 60.3; H, 4.8; N, 2.4.

## 4. Conclusions

Straightforwardly synthesized from the parent dihalophosphines and the corresponding 4-nitrophenols, the terphenylphosphonites ligands **PAr$^{Xyl_2}$(OPh$^{NO_2}$)$_2$** and **PAr$^{Xyl_2}$(OPh$^{NO_2,Me}$)$_2$** feature higher steric demand compared to the related terphenylphosphines, as demonstrated by the steric parameters that we calculated by DFT methods (TCA, %Vbur, %G). The reactions of **PAr$^{Xyl_2}$(OPh$^{NO_2}$)$_2$** with several Pt(II) precursors made evident the high tendency of the ligand to undergo C*sp$^2$*—H activation at an *ortho* position of the **Ph$^{NO_2}$** ring, thus affording five-membered cyclometalated structures in all cases we examined. The enhanced bulkiness and the intrinsic low basicity of ligand **PAr$^{Xyl_2}$(OPh$^{NO_2,Me}$)$_2$** made difficult its coordination to Pt(II) centers via ligand substitution, which could be only achieved using higher temperatures in combination with the organometallic precursor [PtMe$_2$(μ-

SMe2)]$_2$. However, double C$sp^3$—H activation at the methyl groups of the **OPh$^{NO_2,Me}$** moiety was observed in the reaction conditions, with the formation of bis(cyclometalated) structures. The high C≡O stretching frequencies of 2077, 2110, and 2140 cm$^{-1}$ recorded for the carbonyl complexes **1-3·CO**, respectively, are consistent with the expected characteristics of poor σ-donor ability of **PAr$^{Xyl_2}$(OPh$^{NO_2}$)$_2$** and **PAr$^{Xyl_2}$(OPh$^{NO_2,Me}$)$_2$**.

**Supplementary Materials:** The following supporting information can be downloaded at: https://www.mdpi.com/article/10.3390/inorganics10080109/s1, Table S1: Crystal data and structure refinement for compound **1·CO**; Table S2: Crystal data and structure refinement for compound **2**; Table S3: Crystal data and structure refinement for compound **2**.

**Author Contributions:** Conceptualization: R.P. and J.L.-S.; experimental: M.M.A. and M.P.; X-ray diffraction analyses: E.Á.; theoretical calculations: M.M.A. and J.L.-S.; writing—original draft preparation: R.P.; writing—review and editing: R.P. and J.L.-S.; supervision: R.P. and J.L.-S. All authors have read and agreed to the published version of the manuscript.

**Funding:** This research was funded by Ministerio de Ciencia e Innovación (MICINN/AEI/10.13039/501100011033/, "ERDF A way of making Europe", Grant CTQ2017-82893-C2-2-R, Grants CTQ2016-75193-P and PID2019-110856GA-I00), and Junta de Andalucía (Grants US126226 and P18-FR-4688).

**Data Availability Statement:** Not applicable.

**Acknowledgments:** The use of computational facilities at the Supercomputing Centre of Galícia (CESGA) and the Centro de Servicios de Informática y Redes de Comunicaciones (CSIRC), Universidad de Granada are acknowledged. M.M.A. thanks the MICINN for a research fellowship. M.P. thanks the Erasmus Programme for a trainee grant. We also thank E. Carmona for valuable discussion and comments.

**Conflicts of Interest:** The authors declare no conflict of interest.

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
