# Peer review of "New Phosphonite Ligands with High Steric Demand and Low Basicity: Synthesis, Structural Properties and Cyclometalated Complexes of Pt(II)"

_inorganics, doi:10.3390/inorganics10080109_

Round 1

Reviewer 1 Report

Manuscript ID: inorganics-1825292

In the paper titled as “New Phosphonite Ligands with High Steric Demand and Low Basicity: Synthesis, Structural Properties, and Cyclometalated Complexes of Pt(II)”, the authors synthesized two phosphonite ligands containing the highly sterically demanding 2,6-bis(2,6-dimethylphenyl)phenyl group and their corresponding Pt(II) complexes. Additionally, the authors confirmed their structures using different methods.

In general, the Manuscript is very well designed and written. The obtained results are presented and explained in detail. I think that the Manuscript is suitable for the publication in the journal Inorganics in its present form.

Author Response

Thank you for revising our manuscript and for your kind comments.

Reviewer 2 Report

In the presented contribution entitled “New Phosphonite Ligands with High Steric Demand and Low Basicity: Synthesis, Structural Properties, and Cyclometalated Complexes of Pt(II)” authors María M. Alcaide, Matteo Pugliesi, Eleuterio Álvarez, López-Serrano and Riccardo Peloso obtained two phosphonite ligands, which were used for the preparation of the Pt(II) metalorganic complexes and the C-H activation was observed in both systems. Authors supported their results by DFT calculations, multinuclear NMR and X-ray crystallographic studies. In my opinion the presented topic is interesting and can be published in Inorganics after revision. The quality of the presented data is adequate, however, copies of NMR spectra should be included in Supporting Information. The other concerns are listed below:

  1. DFT optimizations were performed for the single molecules of phosphonites showing that the conformations of these systems strongly differ depending on the substitution of OAr aromatic ring. However, it should be kept in mind that the optimization process lead to some local minimum. I advise to perform energy scan around C-P bond with C-C-P-O torsion angle constrained and other parameters optimized during the scan procedure.
  2. Page 3, line 90. This is AA’XX’ spin system, thus observed signals should be interpreted as multipletes, not doublets.
  3. Figures 4 and 5 should be joined.
  4. Experimental section. HH coupling constants should be provided for each multiplete.

Overall due to the above-given reasons, the manuscript can be published in Inorganics MPDI after some revision. Copies of NMR spectra should be obligatory included in SI.

Author Response

1.  We have performed relaxed potential energy scans of the rotation around the P—C bond of PArXyl2(OPhNO2)2 and PArXyl2(OPhNO2,Me)2 at the same level of theory as the geometry optimizations. Also, we have performed full conformational analysis using semiempirical quantum mechanical methods (GFNn–xTB) and the CREST code [[1],[2]]. Analysis of these results confirm the calculated geometries discussed in the main text as the most stable type of conformer for each ligand, even though in the case of PArXyl2(OPhNO2)2  conformations B and C are almost degenerate. In addition, the relaxed scan calculations suggest that P—C rotation is facile for both ligands (almost barrierless for PArXyl2(OPhNO2)2). 

[1] Pracht, P., Bohle, F., Grimme, S.  Phys. Chem. Chem. Phys. 2020, 22, 7169-7192.

[2] Grimme, S. J. Chem. Theory Comput. 2019, 15, 2847-2862.

This text has been added to the supplementary information.

2. Ortho and meta protons of a 1,4-substituted benzene ring rigorously give rise to an AA'XX' spin system in 1H NMR experiment. However, owing to the great difference between the chemical shift separation of the signals (ca 1.5 ppm-600 Hz) and their coupling constant (ca. 8 Hz), this spin syestem appears as an (AX)2 one, i.e. as two well-separated doublets (see the NMR spectra that have been included in the SI). 

3. The two Figures have been joined, as suggested.

4. HH coupling constants are explicitly provided unless they are of the 3JHH type, because all the coupling of this type have very similar values of 7-8 Hz.

Reviewer 3 Report

In this manuscript, the authors reported the synthesis and characterization of two bulky phosphornite ligands and corresponding Pt complexes.  The authors did a great job to describe the reaction procedures experimentally.  I found it is odd that the manuscript does not include a conclusion section.  I would like to recommend to accept this manuscript for publication after adding a conclusion section to summarize the work.

Author Response

Thank you for your comments and suggestions.

A Conclusion Section has been added.

Reviewer 4 Report

Dear Editor,

the manuscript by Peloso et al. reports on the synthesis, structural properties and coordination chemistry of new phosphonite ligands with high steric feature coupled to low basicity.

The work is well-done with a in depth characterization by means of different tools such as DFT, X-ray diffraction and NMR.

Consequently, this Reviewer has nothing to report, except to compliment the authors on the meticulousness of their work and the quality of the presentation of their research.

Consequently, this Reviewer has nothing to report, except to compliment the Authors on the meticulousness of their work and the quality of the presentation of their research. It is therefore with pleasure that I suggest to accept the work in its current form.

Author Response

(The authors gave the same response as above.)

Reviewer 5 Report

The authors described the preparation of three new phosphonite ligands that were characterized by NMR and elemental analysis. Furthermore, the coordination chemistry of the prepared ligands was studied extensively, and some complexes were characterized by single crystal X-ray diffraction. Below are some suggestions for improvement.

1) Wherever needed, the authors could refer to an NMR figure, for example while concluding the symmetry of the ligands from line 82 to 94. Note that the NMR figures could be shown at least in the supporting information

2) Please correct the scheme number in the line 182

3) Please add the reference related to your previous work in the line 232

4) In the discussion part, the authors could highlight the importance of preparing the platinum complexes, especially the CO ones. Knowing that the stabilization of the homoleptic two-coordinate Pt(0) complexes is not discussed.

Author Response

1) NMR spectra have been included in the SI;

2) Corrected;

3) Added;

4) In the first lines of section 2.2 we mention the interesting properties of Pt complexes with terphenylphosphines. On this ground we decided to undertake the study of related complexes with terphenylphosphites, which are also expected to be useful ligands for the preparation of Pt(0) derivatives, as mentioned in the Introduction. Moreover, in the concluding remarks  added comments regarding the carbonyl complexes have been added.

Round 2

Reviewer 2 Report

I have no further comment, the article can be published in the present form.